# The between competition sprint profile of intercounty female gaelic football during training and match-play: An exploratory study

Eddie McGuinness[1,2]*, Mark Lyons[3,4], Kris Beattie[2], Aoife Lane[2], Clement Higginbotham[5], Robin Healy[2]

1 School of Health and Human Performance, Dublin City University, Dublin, Ireland, 2 SHE Research Centre, Department of Sport and Health Sciences, Technological University of the Shannon, Athlone, Ireland, 3 Department of Physical Education and Sports Sciences, University of Limerick, Limerick, Ireland, 4 Sport and Human Performance Research Centre, University of Limerick, Limerick, Ireland, 5 Department of Mechanical, Polymer, and Design, Technological University of the Shannon, Athlone, Ireland

* eddiemc44@gmail.com

## Abstract

Female Gaelic football is an invasive field-based team sport but there are limited data on players' sprint profiles. This study aimed to examine between-competition (league vs championship) differences during training and match-play, and to provide descriptive data by positional group. Fourteen players were monitored using 10-Hz GPS units (STATSports Apex) across 15 games and 34 training sessions. Key metrics included the number of accelerations and decelerations ($\geq \pm 3.0$ m·s$^{-2}$), total sprints ($\geq 20.0$ km·h$^{-1}$), sprint distances (<10 m, 10–20 m, 20–30 m, $\geq 30$ m), average and total sprint duration and distance, peak speed, number of sprints at 80–90% and $\geq 90\%$ of individual peak speed, and the highest percentage of peak speed achieved. In training, average sprint duration and distance, and sprints at 80–90% peak speed decreased from league to championship ($p < 0.05$; ES 0.59–0.84), while peak speed increased ($p < 0.05$; ES 0.61). In games, average sprint duration, number of sprints $\geq 30$ m, and peak speed percentage achieved were greater in league games (p < 0.05; ES 0.36-0.73) compared to championship games. Despite the firmer surfaces of summer championship games, seasonal variations in training focus and in tactical and technical periodisation may explain the observed differences in running performance. These preliminary findings highlight the need for future research to examine how training periodisation, competition standard, and match demands interact to shape the running performance of female Gaelic football players.

## Introduction

Female Gaelic football is one of the most popular female sports in Ireland [1]. The objective is to outscore the opposing team of fifteen players by scoring three points

**Data availability statement:** All data underlying the findings of this study, along with the R scripts used for statistical analyses, are publicly available in the Open Science Framework repository: https://osf.io/f4bdv/overview?view_only=ccbb9a07a08b449c9df90a6507f06c6d. The repository contains the complete dataset and all code necessary to reproduce the results reported in the manuscript. No additional data are required to replicate the study's findings.

**Funding:** The author(s) received no specific funding for this work.

**Competing interests:** NA.

for a goal (under the crossbar and into the net) or one point for a shot over the crossbar and between the uprights. This high-intensity invasive field-based team sport is played over two thirty-minute halves on a rectangular grass pitch (140 x 90 m), and involves a combination of physical, technical, tactical, and psychological components [1–5]. In general, play centres on securing primary possession from the goalkeeper's kickout and transitioning the ball rapidly towards goal to create scoring opportunities. The game emphasises continuous movement, frequent transitions between offence and defence, and a combination of running, soloing the ball, hand passing, and kicking skills. Playing standards in the sport are divided into intercounty and club, with the intercounty competition divided across three playing grades (i.e., senior, intermediate, junior) [6]. Competitively, the season comprises a league competition played in the winter and spring months (January to April), followed by a championship competition in summer (May to August). The championship is traditionally considered the most important competition [7]. Anecdotally, it is suggested that Gaelic football played during the summer months is more entertaining, primarily due to favourable playing conditions (e.g., weather and pitch surface), higher competitive standards, and greater tactical and technical demands [7–9].

Athlete monitoring allows sports scientists to quantify external training loads and assess locomotor performance during competition [10,11]. These data help optimise preparation by guiding training adjustments that reduce the risk of nonfunctional overreaching and subsequent illness or injury [10–12]. External training load monitoring has become popularised over the last decade with the advancement of technology, particularly Global Positioning Systems (GPS) [10,13]. Research conducted in Gaelic games using GPS devices has focused on many elements of training and competition including match-play demands, the influence of different constraints on small-sided game outputs, and the impact of training workload on injury risk [4,14–18]. However, a recent review revealed that over 95% of the research in Gaelic games has been conducted within male cohorts [19].

High-performance female athletes are often underrepresented in sports science literature [19–21]. Sports scientists have a limited pool of research on female athletes and may have to extrapolate data from males or other female sports [6,22]. This limitation raises concerns about generalising findings from men's Gaelic football, given physiological sex differences and the technical and tactical distinctions between the games (e.g., a higher goal frequency and shorter shooting range in the female game) [1,23,24]. Limited data exist on the running demands of female Gaelic football [2,4,19]. Recently, Malone $et$ $al.,$ (2023) reported that, independent of position, players covered a mean total distance of $7319 \pm 1201$ m, a relative distance of $116 \pm 9$ m·min$^{-1}$, a high-speed distance of $1547 \pm 432$ m ($\geq 4.4$ m·s$^{-1}$), and a very high-speed distance of $630 \pm 287$ m ($\geq 5.5$ m·s$^{-1}$) during league, championship and challenge games. The peak speed players attained during match-play was reported as $7.17 \pm 0.41$ m·s$^{-1}$ which represented $86 \pm 4\%$ of maximum sprinting speed [4]. Although the data provided in this study are valuable, an in-depth sprint profile of female Gaelic football remains absent [2,19].

High-intensity activities such as accelerating, decelerating, and sprinting are of vital importance in female Gaelic football [4,21]. Offensively, players who possess greater peak speed are more likely to evade opponents and create scoring opportunities [21]. Defensively, acceleration aids players in a duel for possession and well-developed peak speed allows players to pursue their opponents to make potentially game-winning tackles [15]. From an external workload perspective, while high-intensity movements are linked to a greater risk of injury when monitored effectively, they can also serve as a protective measure against injury [12]. For example, in men's Gaelic football, the appropriate number of exposures to near-maximal peak speed was found to be associated with a reduction in injuries (i.e., 6–10 exposures ≥ 95% peak speed in a 7-day period) [25].

Practitioners require data to effectively monitor and make informed decisions about a team's external workload [26]. Practitioners in female Gaelic football cannot make informed decisions without sufficient data on the high-intensity activity requirements of competition [2]. Consequently, players cannot be optimally prepared for competition demands [10,11,21,22].

Gaelic games research has often collated data from league and championship periods [7]. Therefore, due to the seasonal variation in training and playing conditions, the research produced to date may underestimate the differences in running performance. Firmer summer surfaces and extended training periods may contribute to greater sprint distances and higher peak speeds during the championship compared to league games, as shown in elite senior hurling [7]. These results demonstrate that data on both the league and championship periods are required separately so that practitioners can appropriately periodise their athletes for the demands of the respective competitions. This has yet to be investigated in female Gaelic football. Therefore, the primary aim of the current study was to explore between-competition (i.e., league vs championship) differences in a range of high-intensity activity measures during training and match-play in female Gaelic football with a secondary aim to provide descriptive data on positional groups for games. It is expected that insight into the training and match-play profile in female Gaelic football will aid in advancing and informing future research questions.

## Methods

This was an exploratory observational and descriptive study due to the low sample size involved. A longitudinal design was used to investigate the sprint profiles of elite female Gaelic football players during training and match-play in the league and championship. One intercounty team was monitored during the 2023 season which included seven league (January to April) and seven championship games (April to July) while nineteen league and fifteen championship training sessions were included in the dataset. Training sessions were included in the league and championship periods from seven days before the first game to the last training session of each competition. All games took place between 11:45 am and 4:00 pm. Players followed the team's typical weekly schedule of completing one to two resistance training sessions and two on-field training sessions before playing in the respective competition on the weekend. Eighty-six individual observations were collected during the games (league N = 39; championship N = 47) and 227 training sessions were observed (N = 134 league; N = 93 championship). The average number of observations per player was as follows: championship games (5; range: 2–7), championship training sessions (7; range: 4–15), league games (4; range: 1–7), league training sessions (10; range: 3–13). Game observations were only included in the final analysis if players completed and remained in their starting position for the entirety of the game. Individual observations were further categorised into the player's playing position for a given game. The number of observations was as follows: full-backs (league N = 16; championship N = 16), half-backs (league N = 7; championship N = 10), midfielders (league N = 6; championship N = 8), half-forwards (league N = 9; championship N = 12), and full-forwards (league N = 1; championship N = 1).

### Subjects

Twenty-three players participated in the current study. However, due to the paired design and insufficient data for both league and championship training and games, data for fourteen players (21.7 ± 2.8 years; 1.66 ± 0.04 m; 64.8 ± 9.6 kg)

were retained for analysis. The team participated in Division 3 of the league and in the intermediate championship. Players were selected by the management team to represent their intercounty team as they were deemed the most talented players within that particular county for that season. Consequently, the sample size was constrained and beyond the control of the authors. Ethical approval was granted by the institute's research ethics committee. All participants were informed of the purpose and procedures of the study, and written consent was obtained prior to their participation.

## Experimental procedures

To monitor running performance, players wore 10-Hz GPS devices (STATSports, Apex Team Series, Northern Ireland: firmware version 4.14) with a 100-Hz triaxial accelerometer. The validity of these devices in measuring total distance and peak speed during shuttle run tasks compared to a video camera has been shown to be acceptable [27]. The range of satellites was $20.50 \pm 2.15$ and the horizontal dilution of precision was $0.50 \pm 0.59$ during data collection. Players were assigned and wore a specific unit for the duration of data collection to avoid inter-unit variation similar to previous research [13,19,28]. Players wore the manufacturers' performance vest under their jersey containing the GPS unit in a pouch. GPS devices were turned on a minimum of 30 minutes before the beginning of the game to enable the accumulation of satellite signals (i.e., 'GPS lock') [28]. Before data collection commenced, players were familiarised with the devices by wearing them during training sessions and challenge games for three weeks. The GPS data were downloaded after each game using the proprietary software provided by the manufacturer (STATSports, Apex software version 1.0.0111). This software was not updated during data collection to avoid variation in software-derived data that can occur after a software update [28,29]. Using time stamps, the data were trimmed to only include match-play data (i.e., first and second halves) and training data were trimmed after the warm-up. These data were exported to a customised Microsoft Excel spreadsheet (Microsoft, Redmond, WA, USA) for further analysis.

Before the start of the league, players' maximum sprint speed was calculated during a linear sprint test on an indoor synthetic running track using dual-beam timing gates (Witty, Microgate, Bolzano, Italy) placed at 0, 10, 15, 20, 25, 30, 35 and 40 m. For further detail, please refer to [30] as identical methods were used. Players' maximum sprinting speed was determined so that relative speed thresholds could be calculated from this value [15]. If a player reached a higher peak speed during training or match-play, this new value became their updated peak speed [31]. The relative metrics would then be calculated from this new value. Data collected from the GPS devices included the number of accelerations ($\geq$ 3.0 m·s$^{-2}$), number of decelerations ($\geq$ −3.0 m·s$^{-2}$) [31], total number of sprints ($\geq$ 20.0 km·h$^{-1}$), number of sprints < 10 m, number of sprints between 10–20 m, number of sprints between 20–30 m, number of sprints $\geq$ 30 m, the average duration of a sprint (s), total sprint duration (s), the average sprint distance (m), total sprint distance (m), peak speed (km·h$^{-1}$), the number of sprints at 80–90% of a player's peak speed and the number of sprints $\geq$ 90% of a player's peak speed and the highest percentage of each player's peak speed achieved. The acceleration and deceleration thresholds have been used in elite camogie, international female field hockey and female Gaelic football [4,32,33]. Additionally, the relative speed thresholds have been used in hurling [15]. The minimal effort of duration for a sprint was $\geq$ 1.0 s [15,19,28].

## Statistical analysis

Data and statistical analysis are available in an open online data repository. Normality of all variables was assessed by visually examining the quantile-quantile plots and with Shapiro-Wilk tests with the alpha level set at 0.05. All of the match-play variables were found to be normally distributed except for the average sprint duration, the average sprint distance, the total number of sprints, the maximum percentage of each player's peak speed attained, and the number of sprints $\geq$ 90% of a player's peak speed. For training, the average sprint duration, the average sprint distance, and the number of sprints $\geq$ 90% of a player's peak speed were not normally distributed. All normally distributed variables were described using mean $\pm$ SD while non-normally distributed variables were described using median (IQR).

Paired samples t-tests were conducted to determine whether significant differences existed between league and championship for training and games for normally distributed variables. Cohen's $d_z$ effect sizes were calculated for between-competition differences and interpreted as trivial ($d_z < 0.2$), small ($0.2 \leq d_z < 0.5$), medium ($0.5 \leq d_z < 0.8$) and large ($d_z \geq 0.8$) [34]. Given the exploratory nature of the study, a sensitivity analysis for the difference between two dependent means was conducted using G*Power 3.1. The analysis was performed for a two-tailed test, using a sample size of 14, 80% statistical power, an alpha level of 0.05. The analysis indicated that an effect size (Cohen's $d_z$) of **0.62** could be detected with 80% power and a minimally statistically detectable effect size of 0.43. For non-normally distributed variables, the Wilcoxon Signed Rank Test was used to determine whether significant differences existed between league and championship for training and games. The effect size, r, was calculated using the following equation:

$$r = \frac{z}{\sqrt{N}}$$

where z is the standardised test statistic from the Wilcoxon Signed Rank Test and N is the total number of observations. The r value was interpreted as follows: trivial ($< 0.1$), small ($0.1 \leq 0.29$); medium ($0.3 \leq 0.49$); and large ($> 0.5$) [34]. Statistical analyses were performed using the software RStudio version 2024.12.1+563 (Posit Software, PBC, Boston, MA, USA) with the alpha set to 0.05. Due to the exploratory nature of this study, the alpha level was not adjusted for the number of hypotheses tested.

## Results

Descriptive statistics and between-group effect sizes (ES) are presented for all variables during training and games in Table 1. In training, decreases from the league to the championship were evident in average sprint duration ($p = 0.004$; $r = -0.78$) and average sprint distance ($p = 0.008$; $r = -0.94$), representing large effects. The number of sprints between

**Table 1. Descriptive GPS statistics (mean ± SD or median (IQR)) and effect sizes for training and games between league and championship. Statistically significant differences are highlighted in bold and as asterisk.**

| | Training | | | | Games | | | |
|---|---|---|---|---|---|---|---|---|
| Variable | League | Championship | p | ES | League | Championship | p | ES |
| Accelerations (n) | 30.6 ± 6.1 | 32.2 ± 12.7 | 0.595 | $d_z = 0.15$ | 22.7 ± 7.7 | 23.1 ± 7.4 | 0.858 | $d_z = 0.06$ |
| Decelerations (n) | 27.0 ± 7.1 | 30.8 ± 8.9 | 0.053 | $d_z = 0.57$ | 38.3 ± 13.8 | 34.9 ± 10.0 | 0.134 | $d_z = -0.52$ |
| Average Sprint Duration (s) | 3.7 (0.8) | 3.1 (0.5) | **0.004*** | $r = -0.78$ | 4.0 (0.6) | 3.6 (0.4) | **0.037*** | $r = -0.66$ |
| Total Sprint Duration (s) | 48.9 ± 25.9 | 41.7 ± 22.9 | 0.202 | $d_z = -0.36$ | 62.2 ± 32.1 | 60.2 ± 32.0 | 0.653 | $d_z = -0.15$ |
| Average Sprint Distance (m) | 19.9 (4.2) | 17.5 (3.1) | **0.008*** | $r = -0.84$ | 22.6 (3.4) | 20.1 (3.2) | 0.074 | $r = -0.56$ |
| Total Sprint Distance (m) | 266 ± 141 | 232 ± 129 | 0.248 | $d_z = -0.32$ | 347 ± 185 | 337 ± 186 | 0.673 | $d_z = -0.14$ |
| Peak Speed (km·h⁻¹) | 22.8 ± 0.9 | 23.2 ± 1.2 | **0.041*** | $d_z = 0.61$ | 24.5 ± 1.5 | 24.6 ± 1.5 | 0.234 | $d_z = 0.40$ |
| Sprints (n) | 11.2 ± 4.8 | 11.0 ± 5.8 | 0.836 | $d_z = -0.06$ | 15.4 ± 8.3 | 16.5 ± 8.3 | 0.285 | $d_z = 0.36$ |
| Sprints < 10 m (n) | 2.3 ± 1.1 | 2.8 ± 1.8 | 0.142 | $d_z = 0.42$ | 3.1 ± 2.2 | 3.8 ± 1.9 | 0.198 | $d_z = 0.44$ |
| Sprints 10–20 m (n) | 3.7 ± 1.7 | 4.2 ± 2.4 | 0.239 | $d_z = 0.33$ | 5.3 ± 2.0 | 6.1 ± 3.4 | 0.235 | $d_z = 0.40$ |
| Sprints 20–30 m (n) | 2.4 ± 1.0 | 2.0 ± 1.3 | 0.199 | $d_z = -0.36$ | 3.3 ± 2.2 | 3.7 ± 1.9 | 0.446 | $d_z = 0.25$ |
| Sprints ≥ 30 m (n) | 2.9 ± 2.4 | 2.0 ± 1.8 | 0.054 | $d_z = -0.57$ | 3.8 ± 2.5 | 3.0 ± 2.1 | **0.027*** | $d_z = -0.84$ |
| Highest % of PS Achieved | 86.4 ± 0.3 | 86.1 ± 0.4 | 0.705 | $d_z = -0.10$ | 90.7 (5.2) | 88.8 (4.4) | **0.021*** | $r = -0.73$ |
| Sprints 80–90% PS (n) | 4.9 ± 2.6 | 3.7 ± 2.0 | **0.047*** | $d_z = -0.59$ | 6.0 ± 2.8 | 5.5 ± 2.5 | 0.360 | $d_z = -0.30$ |
| Sprints ≥ 90% PS (n) | 0.5 (0.6) | 0.3 (0.7) | 0.583 | $r = -0.15$ | 0.8 (1.4) | 0.6 (1.0) | 0.241 | $r = -0.37$ |

***Note*** Data presented as mean ± SD or median (interquartile range).

* $p < 0.05$ **ES** Effect size (Cohen's $d_z$ or r), **PS** Peak Speed.

80–90% of a player's peak speed ($p=0.047$; $d_z=0.59$) also declined, indicating a medium effect. Peak speed increased from the league to the championship ($p=0.041$; $d_z=0.61$), representing a medium effect. No statistically significant differences were found for the remaining variables.

Descriptive statistics and between-group effect sizes (ES) are presented for all variables during games in Table 1. During games, the average sprint duration was greater in the league ($p=0.037$; $r=-0.66$), representing a large effect. The number of sprints $\geq 30$ m was also higher in the league ($p=0.027$; $d_z=-0.84$), indicating a large effect. The highest percentage of player's peak speed was greater in league games ($p=0.021$; $r=-0.73$), representing a large effect. No statistically significant differences were found for any of the remaining variables.

Positional data for league and championship games are presented in Table 2 using median and IQR because some positional groups have small sample sizes, providing a more robust report of central tendency and variability. The number of accelerations, decelerations, and sprints performed for training and games during the league and championship are shown in Fig 1. The average peak speed and the percentage of peak speed attained by players in each game during the league and championship are shown in Fig 2.

## Discussion

This exploratory study is the first to examine high-intensity activities during training and match-play in female Gaelic football. Furthermore, this study investigated between-competition differences in high-intensity activities (i.e., league versus championship) in training and games. The primary finding of the current study was that, in both training and games, significantly more sprinting activity was performed during the league compared to the championship, except for peak speed. Additionally, comparative descriptive data indicate a deficit in key variables such as the number of decelerations, sprints, and exposures $\geq 90\%$ of players' peak speed between training sessions and games. It is important to emphasise the exploratory nature of this work; the limited sample size and training adherence percentage and the lack of prior literature,

**Table 2. Descriptive GPS statistics (median (IQR)), for the positional groups during league and championship games.**

| Variable | League Games (N=7) | | | | | Championship Games (N=7) | | | | |
|---|---|---|---|---|---|---|---|---|---|---|
| | FB (N=4) | HB (N=3) | MF (N=2) | HF (N=5) | FF (N=1) | FB (N=5) | HB (N=4) | MF (N=4) | HF (N=4) | FF (N=1) |
| Accelerations (n) | 17.4 (5.4) | 25.7 (10.3) | 29.5 (6.2) | 18.0 (8.0) | 21.0 | 20.0 (11.5) | 24.3 (9.5) | 25.4 (5.2) | 27.5 (7.6) | 19.0 |
| Decelerations (n) | 26.0 (13.1) | 43.0 (3.7) | 50.2 (13.2) | 34.0 (22) | 24.0 | 33.7 (7.3) | 30.7 (8.8) | 34.5 (12.8) | 29.5 (11.2) | 22.0 |
| Average Sprint Duration (s) | 4.0 (0.4) | 3.5 (0.6) | 3.9 (0.2) | 4.1 (1.4) | 3.7 | 3.5 (0.4) | 3.4 (0.8) | 3.6 (0.3) | 3.7 (0.4) | 2.9 |
| Total Sprint Duration (s) | 31.3 (19.6) | 59.4 (10.5) | 107 (20.0) | 85.8 (27.9) | 58.5 | 30.9 (23.7) | 60.5 (23.8) | 97.7 (41.7) | 79.9 (41.9) | 38.3 |
| Average Sprint Distance (m) | 22.2 (3.4) | 19.1 (3.8) | 22.3 (1.4) | 23.2 (6.7) | 20.6 | 19.5 (1.8) | 18.7 (4.9) | 20.5 (2.0) | 21.0 (2.8) | 15.9 |
| Total Sprint Distance (m) | 170 (113) | 324 (61) | 610 (104) | 472 (152) | 329 | 170 (139) | 335 (141) | 561 (254) | 456 (242) | 207 |
| Peak Speed (km·h⁻¹) | 23.2 (1.6) | 24.6 (0.3) | 26.6 (0.9) | 24.9 (0.9) | 26.1 | 24.2 (1.0) | 24.7 (1.2) | 26.2 (0.4) | 25.6 (1.4) | 24.2 |
| Sprints (n) | 7.5 (4.1) | 16.3 (0.5) | 27.7 (6.3) | 17.0 (7.3) | 16.0 | 9 (7.2) | 18 (4.3) | 23.5 (6.9) | 20.5 (11.1) | 13 |
| Sprints < 10 m (n) | 1.3 (1.1) | 2.7 (1) | 6.7 (1.7) | 4.3 (5.0) | 3.0 | 4 (2.6) | 4 (1) | 3 (4.6) | 4.4 (1.7) | 5 |
| Sprints 10-20 m (n) | 3.0 (1.5) | 5.3 (1.3) | 7.0 (2.0) | 5.0 (3.0) | 7.0 | 3.2 (2.2) | 7 (1.9) | 9.3 (2.7) | 7.6 (5.2) | 4 |
| Sprints 20-30 m (n) | 1.2 (0.7) | 4.0 (0.7) | 6.5 (1.8) | 4.0 (3.0) | 3.0 | 2.2 (0.8) | 3.3 (1.6) | 6.1 (2.1) | 3.2 (1.9) | 3 |
| Sprints ≥ 30 m (n) | 1.6 (1.1) | 3.0 (1.7) | 7.5 (0.8) | 5.3 (3.0) | 3.0 | 1.5 (1.3) | 2.7 (1.9) | 3.9 (4.6) | 5.4 (2.1) | 1 |
| Highest % of PS Achieved | 87.6 (1.5) | 90.3 (9.3) | 91.0 (1.3) | 91.0 (1.7) | 96.0 | 90.7 (5.0) | 82.7 (4.1) | 88.4 (6.9) | 90.7 (4.3) | 89.0 |
| Sprints 80-90% PS (n) | 3.2 (2.3) | 6.3 (1.3) | 8.5 (1.5) | 6.0 (3.0) | 7.0 | 5.2 (1.2) | 2.0 (3) | 6.0 (2.6) | 7.0 (4.3) | 5 |
| Sprints ≥ 90% PS (n) | 0.3 (0.5) | 0.7 (4.8) | 0.8 (0.2) | 1.3 (2.3) | 1.0 | 1.0 (0.5) | 0 (0.5) | 0.6 (1.4) | 1.1 (1.8) | 0 |

**Note:** N equals the number of players contributing observations to that position. To avoid overweighting players with multiple games, we first calculated each player's mean for a given position (player-position average). Positional summary statistics (median and IQR) were then computed across those player-position averages. **FB** Full-back, **FF** Full-forward, **HB** Half-back, **HF** Half-forward, **MF** Midfield, **PS** Peak Speed

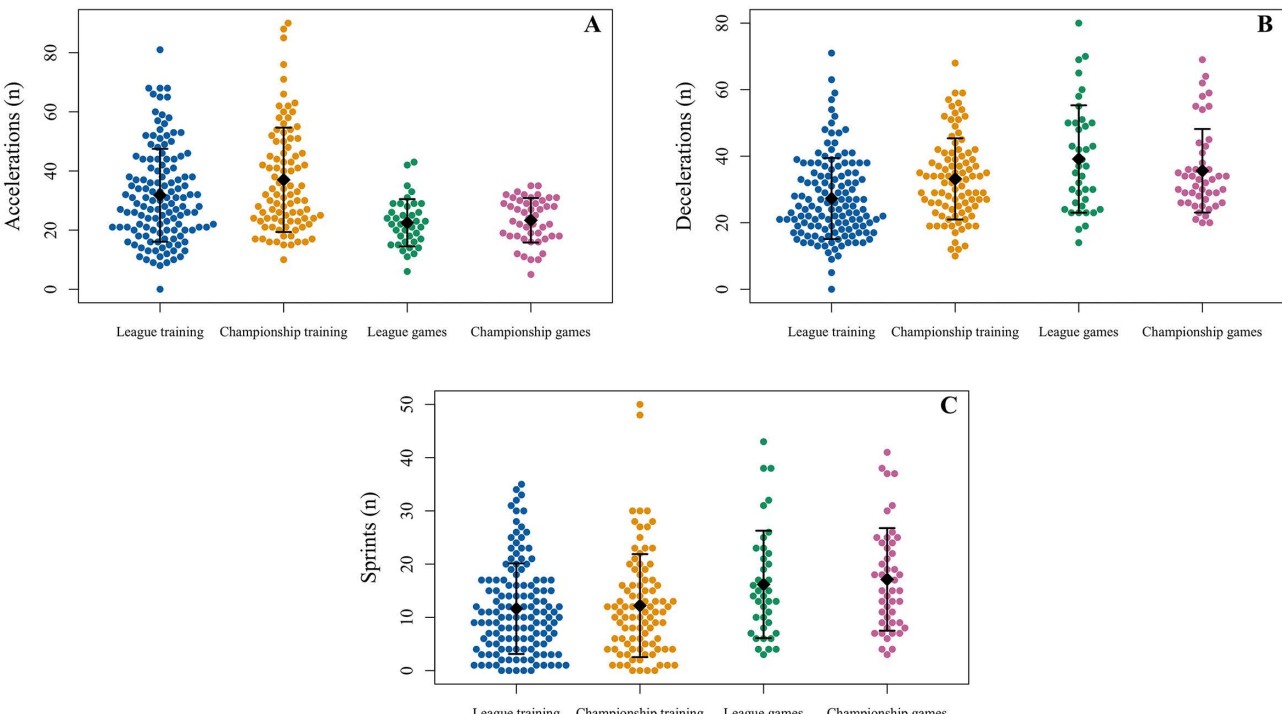

**Fig 1. Observations for the number of accelerations (≥ 3 m·s⁻²) (A), decelerations (≥ −3 m·s⁻²) (B), and the number of sprints (≥ 20 km·h⁻¹) (C) performed for training and games during the league and championship.** Each point represents a single-player session. Mean values are shown with diamond markers, and error bars represent the standard deviation.

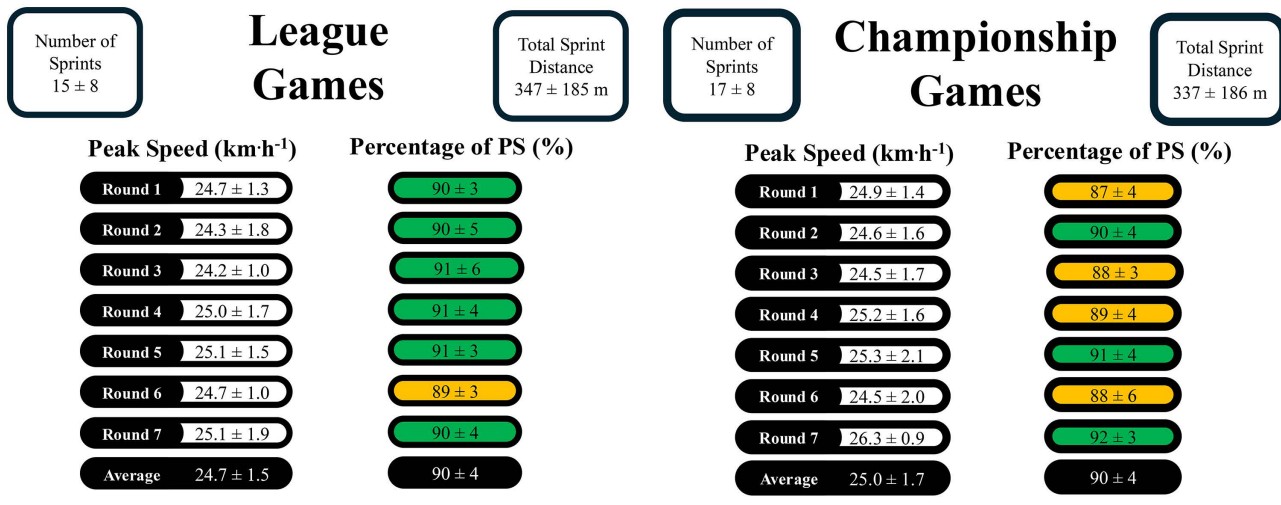

**Fig 2. Peak speed and the percentage of peak speed attained.** Reported per game for league and championship.

made it difficult for the authors to draw strong conclusions from the collected data. Further confirmatory investigations are necessary to confirm the current findings.

The physical performance of team sport athletes can change significantly throughout a season (i.e., speed, power, and strength), as well as their activity profile during training or match play [35–38]. Peak speed during training increased significantly from league to championship (medium effect) whereas the average sprint duration and distances and the number of sprints performed between 80–90% of players' peak speed significantly decreased from the league to the championship (large effects). The observed increase in peak speed may be partly attributable to the firmer playing surfaces typical of summer [8]. This also likely reflects a positive adaptation to sprint training performed throughout the year, including both general- (e.g., strength training) and specific- (e.g., free sprinting) methods. The specific content of each training session was not objectively recorded which is acknowledged as a limitation to the current study. However, overall trends in training direction by the coaching staff as the season progressed may somewhat explain some of the reduced sprint activity. For example, the team anecdotally spent more time completing conditioning work, and as the season developed and the team were deemed to have reached appropriate levels of aerobic and anaerobic endurance, this time was reallocated to technical/tactical activities. These activities typically include technical work such as passing and shooting drills, game-based scenarios, practicing restarts, and adjusting the team formation in anticipation for future opposition. Drills involving technical/tactical activities have been shown to produce significantly less sprint distance than conditioning [26]. Additionally, other measures of training load may have influenced the adjustment of training content (e.g., heart-rate metrics, ratings of perceived exertion, athlete self-reported measures, player feedback etc.) [10,11,39]. In female soccer, Mara *et al*., [37] observed significant decreases in sprint distance as the season progressed, which were attributed to a more condensed schedule later in the season. This unlikely explains the current study's findings as the recovery times between games in the league and championship competitions were similar. The data reported highlight the importance of balancing physical and tactical preparation. While players achieved higher peak speeds later in the season, reduced sprint frequency suggests lower exposure to repeated high-intensity actions. Technical coaches and sport scientists should work collaboratively to monitor sprint exposure while periodising technical and tactical priorities. Incorporating an appropriate volume of sprints into skill-based drills can help maintain neuromuscular adaptations and ensure readiness for the repeated high-intensity efforts demanded during championship competition.

In games, similar trends were found with significant decreases from league to championship for average sprint duration, the number of sprints ≥ 30 m, and the highest percentage of a player's peak speed. These decreases suggest that players are required to perform sprints of reduced duration and distance, at lower relative speeds, during championship games. Several reasons may help explain these observed changes. Championship games are typically 'faster-paced', influenced by higher fitness levels, greater tactical awareness and organisation, enhanced skill execution, and improved playing conditions. Increased fitness allows players to apply more consistent defensive pressure (i.e., execute tackles, cover space) while stronger tactical structures enable teams to limit time and space on the ball more effectively. As a result, players in possession have fewer opportunities to solo the ball over extended distances, and defenders are required to engage in fewer long-distance pursuits. Improved skill execution and more favourable summer conditions also tend to reduce the number of turnovers. With fewer turnovers, defensive and offensive transitions are less frequent, which are the moments most likely to elicit high-speed running and longer sprint efforts. Collectively, these factors reduce the frequency and necessity of long-distance sprinting and diminish the opportunities for players to reach higher relative speeds during championship games. Research in female Gaelic football has found that 26% of players require 30–40 m to reach their maximum sprinting speed [30]. With fewer long sprints, it follows that the average sprint duration also declined. Other factors, such as playing style and opponent ranking, can also affect match-play running performance in team sports [1,26,40,41]; these determinants have not yet been investigated in female Gaelic football. Understanding how these contextual variables interact with competitive phase would offer important insights into the tactical and physical demands in games and should be examined in future research.

To gain a deeper understanding of match-play running performance, high-intensity activities must be taken into account [11]. High mechanical load activities such as accelerations, decelerations, and sprinting, are linked to greater injury risk [42,43]. These activities also impact performance; for example, faster peak speeds can distinguish between higher- and lower- calibre players, and higher playing standards involve more accelerations and decelerations [18,44,45]. In this current study, players performed $23 \pm 7$ accelerations during championship games, whereas work from Malone et al. [4] reported that players performed $42 \pm 6$ accelerations per game. Similarly, peak speed, an essential physical quality to possess for evasion and pursuit, was reported as $25.8 \pm 1.5$ km·h$^{-1}$ which is nearly 5% higher than the $24.6 \pm 1.5$ km·h$^{-1}$ reported in the current study [4]. In elite camogie, peak speeds of $25.4 \pm 1.5$, $24.9 \pm 1.6$ and a median of 26.0 km·h$^{-1}$ have been reported during match-play [26,32,46]. The disparity in values in female Gaelic football is likely due to the teams competing at different levels (i.e., senior vs. intermediate) as well as methodological considerations (e.g., brand/model of GPS device, satellite signals, interunit reliability) when using GPS technology. The higher-calibre team in Malone et al. may also possess superior neuromuscular and physical qualities, enabling players to perform more accelerations and attain higher peak speeds [47]. From an applied perspective, these findings highlight the need for practitioners to consider competitive standard and data-collection methods when benchmarking running performance. Comparison of intermediate-level to senior-level players should be done cautiously. A recent study in female Gaelic football found that Inter-county players demonstrated significantly greater peak force, jump height, and reactive strength index than club-level players [6]. Such differences in underlying neuromuscular qualities may partly explain why players at higher standards achieve more frequent accelerations and reach greater peak speeds than intermediate-level counterparts. Intermediate-level coaches should therefore focus on developing technical and neuromuscular qualities that support frequent accelerations and high peak speeds (e.g., explosive- and reactive- strength) alongside appropriate sprint volume and speed training.

Understanding the occurrence of deceleration during match-play is critical given its association with high mechanical load and catastrophic injuries, particularly in female athletes [48]. Similar to research in male team sports and camogie as well as considering data from the current study and others, decelerations appear to occur more frequently than accelerations in elite female Gaelic football (decelerations n = 38–53; accelerations n = 23–42) [4,13,26]. It is prudent, therefore, that future injury prevention programs contain elements to enhance deceleration-specific tissue properties and coordination given the number of decelerations and the association with knee injuries in female Gaelic games players [42,48–50].

There is a disparity between the volume of sprint distance performed in the current study in comparison to other work in elite female Gaelic football ($630 \pm 287$ vs $347 \pm 185$ m) [4]. Our data are similar to those reported in female international soccer ($364 \pm 53$ m) yet markedly greater than results found in camogie (160–256 m) [26,32,45,46]. The possible greater sprint demand in female Gaelic football compared to camogie is similarly reflected in male athletes. Young et al., [15] reported a 9% greater total match-play sprint distance in men's Gaelic football when compared to hurling. In Gaelic football, players often need to solo the ball and support teammates in possession. In camogie and hurling, a single pass can send the ball 60–80 m, likely reducing the sprinting demand [19].

Exposure to peak- or near- peak speeds has become topical with research suggesting that an optimal dose of exposures above a certain threshold (i.e., > 90–95%) of players' peak speed can have injury reduction benefits in Gaelic football [25,51]. Furthermore, the principle of specificity also suggests that frequent exposure to high speeds enhances maximum sprinting speed [52]. However, our data highlight that players may not be exposed to sufficient efforts in training during the league and championship periods. As a potential consequence, the number of sprints ≥ 90% in games is limited. For example, in games, players in the current study completed noticeably fewer sprints ≥ 90% of their peak speed than reported values in elite camogie ($3 \pm 3$ sprints) and hurling ($3 \pm 2$ sprints) [15,32]. The percentage of peak speed attained in games (89–91%) falls between values found in female Gaelic football ($86 \pm 4\%$) and elite hockey ($91 \pm 4\%$) [4,31], high-lighting an evidence-practice gap between the recommendation of regular sprint exposure (1–3 efforts per week ≥ 90% peak speed) and actual training practices [2,51]. This underexposure may limit players' ability to reach near-peak speeds during games and increases the risk of hamstring strain injuries [25,51]. Examining the training and match-play data

further reveals some mismatches: total sprint distance and decelerations were lower in training than in games, suggesting players may be underprepared for match demands [53,54]. Progressive, game-replicating sprint and deceleration training is therefore essential to prepare athletes physically for competition, enhance sprint capacity, and reduce injury risk [48,51,53,55]. In collegiate female Gaelic football, hamstring injuries account for 22% of total injuries and sprinting is a common mechanism of injury [50,56]. Accordingly, practitioners should prioritise providing players with sufficient volumes of near-peak sprint exposure in training to optimise match-play performance and minimise the risk of hamstring injury [25,51]. Although general guidelines recommend 1–3 exposures per week at ≥ 90% of peak speed, these targets should be contextualised within the specific demands of Gaelic football and tailored to individual players, positional roles, and weekly training loads.

'High-performance' female athletes are underrepresented in sports science literature [20]. The lack of travel expenses, gym facilities, strength and conditioning staff, and appropriate training pitches apparent in elite female Gaelic football has been reported [57]. These factors may have led to low training attendance in this study, with players attending only 50% of league training sessions on average. Inconsistent attendance posed a major problem, making it difficult to draw strong conclusions from the limited data. In the last twenty years, only four teams have won the Senior All-Ireland which suggests that there is a substantial gap between the All-Ireland contenders and the remaining teams. Future research should examine attitudes, beliefs, and barriers to training attendance across all levels of intercounty female Gaelic football. This study's findings are confined to one team, limiting generalisability. The low attendance could be an isolated issue, indicating a lack of player 'buy-in' for this particular team. However, small sample sizes are a common limitation in sports science literature [58]. Our study supports the idea of data sharing and pooling among researchers to increase sample size, statistical power, the precision of estimated effects and enhance the generalisability of findings by reducing the influence of individual team characteristics.

## Limitations

Several limitations should be acknowledged. The small sample size and reduced training adherence limited the statistical power and generalisability of the findings; however, the study was exploratory and aimed to generate informed hypotheses for future research. As the investigation focused on a single intermediate-level team, the results may not fully represent Gaelic football teams training and competing at different competition levels with difference resources or training structures. The absence of detailed session content restricted the ability to interpret observed differences in training and match demands. Additionally, while multiple comparisons were performed without correction, effect sizes and confidence intervals were reported to aid interpretation of practical relevance. Finally, although GPS technology carries inherent measurement error, the same device model and consistent data-processing procedures were used throughout to enhance reliability.

## Future directions

Future research should aim to include larger, multi-team cohorts across different competitive levels to enhance statistical power and generalisability. Collaboration and data sharing among researchers could facilitate this, enabling more robust analyses of positional and contextual influences on running performance. Further work should also incorporate detailed assessments of training content to better explain observed seasonal variations in high-intensity activities. Qualitative research examining player and coach perspectives on training attendance, barriers, and facilitators could help address the training attendance patterns observed in this research.

Based on the patterns observed in the present exploratory analysis, several testable hypotheses emerge for future confirmatory research. Such studies might investigate whether decreases in average sprint duration and distance from league to championship replicate in larger samples; whether shifts in sprint-intensity distribution (fewer but faster sprints) are consistent across competitions and generalise to match play; and whether sprint volume and exposures ≥ 90% of players'

peak speed influence non-contact injury risk. Confirmatory testing of such hypotheses would contribute to improved understanding of sprint-load management and injury prevention in female intercounty Gaelic football.

## Conclusions

The current study was one of the first to report data for the high-intensity activities in elite female Gaelic football during games and training and compare these data between the league and championship competitions. This exploratory work suggests that excluding peak speed, the average sprint duration and distance, and the number of sprints between 80–90% of players' peak speed were significantly reduced from league to championship training. In games, significant reductions from league to the championship in the average sprint duration, and the number of sprints ≥ 90% of players' peak speed and percentage of peak speed attained were observed. An evidence-practice gap seems apparent with the lack of exposures ≥ 90% of players peak speed in training; potentially increasing the risk of injury. Furthermore, the discernible decrease between the number of decelerations and the total sprint distance performed in training compared to games suggests that players are potentially underprepared for games. Thus, the match-play data may not truly represent the match-play demands but rather the sprint performance of the current cohort.

## Author contributions

**Conceptualization:** Eddie McGuinness, Mark Lyons, Kris Beattie, Robin Healy.

**Data curation:** Eddie McGuinness, Robin Healy.

**Formal analysis:** Eddie McGuinness, Robin Healy.

**Investigation:** Eddie McGuinness.

**Methodology:** Eddie McGuinness, Robin Healy.

**Software:** Eddie McGuinness.

**Supervision:** Mark Lyons, Kris Beattie, Aoife Lane, Clement Higginbotham, Robin Healy.

**Visualization:** Eddie McGuinness.

**Writing – original draft:** Eddie McGuinness.

**Writing – review & editing:** Eddie McGuinness, Mark Lyons, Kris Beattie, Aoife Lane, Robin Healy.

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
