## [Decision Letter · Decision Letter 0]

20 Oct 2025

Dear Dr. McGuinness,

Thank you for submitting your manuscript to PLOS ONE. After careful consideration, we feel that it has merit but does not fully meet PLOS ONE’s publication criteria as it currently stands. Therefore, we invite you to submit a revised version of the manuscript that addresses the points raised during the review process.

I have attached the academic editor’s review and an additional review from a respected academic and practitioner within Gaelic sports. Please revise the manuscript in accordance with the feedback provided in both reviews.

Each comment should be addressed in detail, and a corresponding response should be included for every point raised in both the editorial and peer review reports.

Thank you for your careful attention to these revisions prior to resubmission

We look forward to receiving your revised manuscript.

Kind regards,

Shane Malone, Ph.D.

Academic Editor

PLOS ONE

Journal Requirements:

2. Thank you for stating the following in your Competing Interests section:  “NA”

4. We note that Figure 2 in your submission contain copyrighted images. All PLOS content is published under the Creative Commons Attribution License (CC BY 4.0), which means that the manuscript, images, and Supporting Information files will be freely available online, and any third party is permitted to access, download, copy, distribute, and use these materials in any way, even commercially, with proper attribution. For more information, see our copyright guidelines: http://journals.plos.org/plosone/s/licenses-and-copyright.

1)  You may seek permission from the original copyright holder of Figure 2 to publish the content specifically under the CC BY 4.0 license.

2) If you are unable to obtain permission from the original copyright holder to publish these figures under the CC BY 4.0 license or if the copyright holder’s requirements are incompatible with the CC BY 4.0 license, please either i) remove the figure or ii) supply a replacement figure that complies with the CC BY 4.0 license. Please check copyright information on all replacement figures and update the figure caption with source information.

If applicable, please specify in the figure caption text when a figure is similar but not identical to the original image and is therefore for illustrative purposes only.

**Additional Editor Comments:**

“The Between Competition Sprint Profile of Elite Female Gaelic Football during Training and Match-Play: An Exploratory Study”

Editor General Assessment

I would like to thank the authors for the opportunity to act as the editor for the above piece. The current paper addresses an important gap in sports science literature within Gaelic Football by examining sprint profiles in elite female Gaelic Football across league and championship phases. Which has important implications for the training process within the game. The use of GPS-derived data in female athletes is novel and valuable. However, there are several areas where the manuscript would benefit from refinement:

1. Language and grammar polishing to improve readability and flow.

2. Punctuation consistency in reference citations, abbreviations, and statistical reporting.

3. Scientific rigor improvements relating to sample size justification, interpretation of effect sizes, and the reporting of methodological limitation

Major Comments

1. Sample Size and Generalisability

o Only 14 players’ data were retained for analysis out of 23, representing a single Division 3 team. This severely limits generalisability to elite female Gaelic football as a whole. It is important to understand this when discussing your findings across the manuscript

o The exploratory and descriptive nature is acknowledged, but the paper should emphasise that conclusions are preliminary and hypothesis-generating rather than having a significant impact, it is important to not over emphasise findings, and place them appropriately within the context of the research

investigation completed.

2. Effect Sizes and Statistical Testing

o Multiple comparisons were performed without correction (alpha not adjusted). While this is acknowledged due to the exploratory design, readers should be cautioned more strongly about inflated Type I error.

o At times, non-significant findings with “approaching significance” are over-interpreted. These should be discussed cautiously.

3. Training Session Context Missing

o The manuscript admits that training content was not recorded. This is a major limitation: without knowing whether training focused on technical, tactical vs conditioning drills, these are common within championship and naturally decrease the running demands of training, as there is a focus on tactical

clarity and freshness leading into more important games, it is important to acknowledge this when interpreting sprint and acceleration data, without this, your elucidations are very speculative.

4. Use of GPS Technology

o While validity of 10-Hz GPS is noted, differences in device accuracy compared to other studies are not fully considered when comparing values across sports, include as limitation or provide additional support for validity and reliability

5. Terminology and Definitions

o Terms such as “elite” may need to be clarified given that the cohort was from Division 3 and the intermediate championship. More precise terminology (e.g., “inter-county intermediate-level players”) would enhance clarity.

Minor Language, Grammar, and Punctuation Issues

Abstract

• Line 32: “…with limited data on players’ sprint profiles.” Replace “with” but limited data exist on players’ sprint profiles.” for clarity.

• Line 33: “and provide descriptive data by positional group.” → Better phrased as “…and to provide descriptive data by positional group.”

• Line 35: “…across 15 games and 34 training sessions.”

Sample size is relatively small; consider flagging this as a limitation within manuscript limtations and further research section within discussion

• Line 41: “…significantly greater in league play (large effects, p < 0.05).” The phrase “significantly greater… (large effects, p < 0.05)” risks overstating exploratory data; note just large effects were observed between league and championship, given low sample size.

• Line 42: “…unexpected given the firmer surfaces typical of summer championship games.” Add nuance: surface conditions may not be the only explanatory factor (coaching, tactical demands, focus on freshness over fatigue during championship, importance of competition etc.) Not all the variance is explained by surface, oversimplification of your findings and has very little coaching value, what do your findings mean for the coaches on the ground.

Introduction

• Line 47: “…dynamic requiring players…” Insert comma: “…dynamic, requiring players…”

• Line 55-59: “….Athlete monitoring enables the sports science practitioner to determine athletes’ internal and external responses to training loads and match-play (7,8). These data can allow practitioners to prepare ptimally for competition by planning and adjusting training loads to reduce the risk of non-functional over-reaching which may result in an increased risk of illness or injury (Kelly et al., 2022; O’Connor et 58 al., 2021; (7–9)….” As your research is focused on external load and locomotor patterns of the game, there is no need to reference internal loading, I would suggest a restructure of this paragraph

• Line 57: “…non-functional over-reaching which may result…” Hyphenation is not required: “nonfunctional overreaching, which may result…”

• Lines 58–60: Delete Kelly et al., 2022; O’Connor et al., 2021 and keep reference number (7–9).”

• Line 63: “…over 95% of the research in Gaelic games has been conducted in the male population (16).” Suggest rephrasing: “within male cohorts”

• Lines 66–67: Repetition of “consequently” in two consecutive sentences, need to improve the flow of your sentence construction here.

• Line 99: “with a secondary aim was to provide…” Grammar issue here: “with a secondary aim to provide”

Methods

• Line 134: “…written consent was obtained prior to their participation..” Double period remove one.

• Line 138-139: “….The validity of these devices in measuring total distance and peak speed during shuttle run tasks compared to a video camera has been shown to be acceptable…..” – Provide the validity and reliability of the specific metrics of Accel, Decel and sprint demarcations used within your investigation, if these can not be found provide data from within your research group on the validity and reliability of these data.

• Line 145: “……players were familiarised with the devices by wearing them during training sessions and challenge games three weeks…..” Revise to “for three weeks”

• Line 156: “…..For further detail, please refer to (cross-156 sectional ref)….” ….Include the reference here !

• Line 158–159: “…this new value became their updated peak speed. (29).” Extra period before citation, remove.

• Line 165: “…and the highest percentage of a peak speed player’s achieved.” This is very clunky phrasing; suggest you revise to “…and the highest percentage of each player’s peak speed achieved.”

• Line 169: “……relative speed thresholds have been used in hurling (12,30,31). The minimal effort of duration for a 168 sprint was ≥ 1.0 s 23/09/2025 06:25:00….”…..Remove the date here which appears to be a inserted comment from a fellow authorship member

Results

• Line 197-210: Include the ES number, the inference and then the P-Value, this provides clarity to all your findings. I also feel you should provide an introduction to the overall results in mean and SD and positional differences and then get into the differences between competitions. I think you need to be careful with the use of statistically significant, given small sample sizes.

Discussion

• “The lack of significance in combination with the low sample size constrain inferences…” → Should be “constrains inferences…”

• “None of these factors have yet to be investigated…” Better phrased as “…have been investigated…”

• “Since sprinting is a common cause of injury in female collegiate athletes and hamstring injuries account for 22% of injuries in collegiate female Gaelic football…” This is a long sentence and could be split for readability.

• The authors need to add an appropriate limitations and future research section at the end of the discussion

• An overall comment would be that the authors need to do more than state the findings, answer always for each paragraph, what do our findings mean for players and coaches of the sport, how will the data aid training construction and within match-play performance decisions from coaching and performance staff

References

• Inconsistent formatting of references throughout the paper, please complete review in line with journal requirements.

• Inconsistent formatting for references and names and punctuation (e.g., some include full stops after abbreviations, others do not).

• Ensure all cross-referenced “(cross-sectional ref)” and “(survey ref)” placeholders are replaced with complete citations.

Tables and Figures

Table 1

• Inconsistent p-value spacing (p <0.05 vs p < 0.05).

• Missing ± SD for specific metrics that need to be added

• Suggest highlighting in Bold the effect size that met the G-Power threshold for 80% Power

• Abbreviation key (PS = Peak Speed) should be moved to the table caption for clarity.

Table 2

• Missing ± SD for specific metrics that need to be added

Figure 1

• Suggest making bar whisker plots and keep the individual bee swam at the back of the plot, allows for more understanding for wider sports science community.

• Axis labels not shown in text snippet — ensure units (e.g., “Number of sprints [n]”) are clearly labeled.

• Caption could better explain the plot used and how data are presented.

Figure 2

• This is an infographic, and while it is cool, it needs more scientific rigor applied, I suggest adding mean and SD to the figure and rounding the %Peak Speed to a round figure with SD

• Caption: “ indicates a median value”* Asterisk notation unclear; should specify “ indicates statistically significant difference (p < 0.05)”* if that is the case.

• Abbreviation (PS = Peak Speed) repeated but should be defined once.

Editorial Recommendation

Major Revisions but can be published with PLOSone post edits and reviewer revisions are completed.

The study has merit and novelty but requires:

• Stronger acknowledgment of limitations in sample size, training context, and device comparisons.

• Careful moderation of conclusions to reflect exploratory rather than confirmatory findings.

• Polishing of grammar, punctuation, and reference formatting for increased readability.

Reviewers' comments:

Reviewer's Responses to Questions

**Comments to the Author**

1. Is the manuscript technically sound, and do the data support the conclusions?

Reviewer #1: Partly

2. Has the statistical analysis been performed appropriately and rigorously?

Reviewer #1: Yes

3. Have the authors made all data underlying the findings in their manuscript fully available?

Reviewer #1: Yes

4. Is the manuscript presented in an intelligible fashion and written in standard English?

Reviewer #1: No

Reviewer #1: Thank you for the opportunity to review this paper. This is a potentially interesting paper. This area of female sports is under researched and as a result, this information could add the field. However, there are major revisions needed as the paper is tangential in places and doesn’t critically analyze the results through the discussion. The submission seems incomplete; there are errors throughout which should have been corrected pre-submission.

General feedback:

Please complete a through proof-read before re-submission. Improve writing clarity — several grammatical errors and awkward phrases reduce readability. The paper could have potential value and could fits the journal’s scope, but significant methodological and reporting issues must be addressed. The authors should refine their introduction discussion before the manuscript can be reconsidered. Your findings need to be expanded in line with best sports science practices in elite sports. There is an opportunity here for you to provided recommendations for coaches/sports scientists to replicate the demands of the game in training to ensure the players are best prepared for competition. Check reference style to ensure compliance with journal guidelines.

Specific feedback:

Abstract:

Line 34: Why only 14 players?

Line 40: Should specific effect size be reported here (p < 0.05, ES = 1.2)? p should also be in italics throughout the document. Please amend throughout the document.

Line 42-43: Sentence need to be re-written.

Line 43: There is always further research needed, could more specific recommendation be provided here based on what your study found?

Introduction:

Needs to be proof-read and re-formatted throughout the section.

Line 47-49: I get what you and trying to say here, re-write and re-phrase.

Line 47-54: I would suggest a little more context around the unique principles of play, and the unique formatting and structure of the NL and AIC in relation to LGF. Journal will have international readership so a greater outline of the game in needed here.

Line 52: ‘suggested’ instead of ‘purported’, doesn’t read well.

Line 52-54: references at the end a sentence (5, 6).

Egan et al., is from intercounty hurling research. I would suggested to use read the following papers which looked at some of the contextual demands in male GF:

Mangan, S., Malone, S., Ryan, M., McGahan, J., O’Neill, C., Burns, C., Warne, J., Martin, D & Collins, K. (2017). The influence of match outcome on running performance in elite Gaelic football. Science in Medicine & Football 1, 272-279.

Mangan, S., Ryan, M., Shovlin, A., McGahan, J., Malone, S., O'Neill, C., Burns, C., & Collins, K. (2019). Seasonal changes in Gaelic football match-play running performance. Journal of Strength & Conditioning Research 33, 1685-1691.

McGahan, J., Mangan, S., Collins, K., Burns, C., Gabbett, T., & O’Neill, C. (2021). Match-play running demands and technical performance among elite Gaelic footballers: Does divisional status count? Journal of Strength & Conditioning Research 24, 169-175.

Additionally, these can support/refute your findings in the discussion.

Line 55: remove ‘the’, add sports science practitioners.

Line 56: This data….. delete These.

Line 58: Delete Kelly et al., 2022; O’Connor et al., 2021; (7–9). Ensure in-text referencing is consistent and in line with journal recommended referencing style.

Has this paper been submitted elsewhere?

Lines 61-63: Change this statement to the following:

Previous research conducted in Gaelic games using GPS technology has focused on many elements of training and competition including match-play demands, the influence of different constraints on small-sided game outputs, and the influence of training workload on injury risk (2,11–15).

Line 65: This is true but there is specific female GAA research to support your statement here. References 16 & 22.

Lines 66-68, Consequently, used twice in four sentences.

Lines 70-71: Re-write, suggested:

There is currently a dearth of research conducted in intercounty female Gaelic football….

Line 71: et al., should be in italics, with the inclusion of a comma.

Line 71-74 during competitive games? This is mis-leading as the research was conducted over two seasons with no differentiation between NL or AIC.

Lines 76-78: Please re-write and re-structure.

Line 81: Would the Ladies GF recommendations paper be a more appropriate reference here, to compare apples with apples.

Line 82: Reference 12 = hurling research. Very different physical, technical & tactical demands in comparison to GF. I would recommend making comparisons to LGF as much as possible and then comparison it to the male version of the game.

Please amend/adjust throughout the document.

Line 86-87: Good example of GF comparison here.

Line 94-96: In line with previous comments, are there similarities in Gaelic football.

Line 99: Ladies or female Gaelic football, keep consistent throughout the document, intercounty would be have to specified here also.

Lines 99-102: Does the journal require a hypothesis statement? Editor can advise.

Methodology

This section is generally well written.

Experimental Procedures

Line 137 – The pro series has been validated as the team series yet? If not, needs to be included in the limitations section.

Line 156-157 – ‘please refer to (cross sectional ref) as identical methods were used.’ What is this referring to?

Line 157-158 – Please refer to the following papers for guidance on the utilization of relative & absolute speed thresholds.

Could be of use in the discussion section:

Gualtieri, A., Rampinini, E., Dello Iacono, A., & Beato, M. (2023). High-speed running and sprinting in professional adult soccer: Current thresholds definition, match demands and training strategies. A systematic review. Frontiers of Sports Activity and Living 5, doi: 10.3389/fspor.2023.1116293.

Park, L.A.F., Scott, D., & Lovell, R. (2019). Velocity zone classification in elite women’s football: where do we draw the lines? Science & Medicine in Football 3, 21-28.

Line 166-168 – are there any Gaelic Football relative speed thresholds available.

Lines 166-168 – slight re-structuring and split sentence into two:

The absolute speed, acceleration and deceleration thresholds have been used previously in intercounty camogie and international female field hockey. Additionally, the relative speed thresholds have been used in intercounty hurling (12,30,31).

Did this paper not use similar velocity thresholds?

O’Grady M, Young D, Collins K, Keane J, Malone S, Coratella G. An Investigation of the Sprint Performance of Senior Elite Camogie Players during Competitive Play. Sport Sci Health. 2022 Jan 16.

Line 169- 23/09/2025 06:25:00, please delete.

Statistical Analysis

Well written, clear and transparent, repeatable.

Results

Line 200 – p value first, ES as in abstract?

Young D, Malone S, Collins K, Mourot L, Beato M, Coratella G (2019) Metabolic power in hurling with respect to position and halves of match-play. PLoS ONE 14(12): e0225947. https:// doi.org/10.1371/journal.pone.0225947.

This paper may be of use for structure and formatting purposes.

The result section is very brief,

The descriptive statistics and effect sizes for training and games between league and championship appears to be missing?

Line 212 - In table 1, the championship total sprint distance is 337 ± 339, is the SD correct here?

Lines 220 & 224 – Good visualization of the data here.

Discussion

General Feedback

The discussion should better integrate findings with existing literature. Currently, it reads more like a summary than a critical evaluation. Clearly distinguish between statistically significant results and practical applications for intercounty LGF players.

Can recommendations be made regarding the minimal effective dosage of specific physical qualities in line with best practice with other team sports?

Just a suggestion possible utilization of Tactical Periodization to marry physical, technical and tactical as per recommend in Gaelic Football:

Timmons, K., Collins, K., & Mangan, S. (2025). The science of Gaelic football match-play and recommendations for future research directions: a narrative review. Sport Sciences for Health, 1-17.

Mangan, S., Collins, K., Burns, C., & O’Neill, C. (2021). A tactical periodisation model for Gaelic football. International Journal of Sports Science & Coaching, 17(1), 208-219. https://doi.org/10.1177/17479541211016269.

Please check the correct reporting of ES in the discussion as per journal guidelines.

Line 230- intercounty, please keep terminology consistent throughout the document.

Line 230- Furthermore

Line 232 – performed rather than recorded?

Line 235-238 - Not sure this is appropriately placed here.

Line 252 – Research conducted in elite female soccer….(35). Research at the end of sentence to enhance flow.

Line 259-261 – very repetitive here, see feedback regarding best practices in other sports and it can be implements in LGF.

Line 265 – Shorter sprint burst, colloquial, consider the international readership of the journal.

Line 267 – first mention of solo, again reader may be unaware of what this specific skill is, please outline in introduction.

Line 268-269 - MSS should still be exposed their MSS (vaccine) as you have previously eluded to in the introduction.

Line 270 (MSS, ref)

Line 270-271 – Poor sentence, re-write.

Lines 277-279 – These actions are also linked to gal scoring opportunities in other sports.

Line 282- incomplete reference

Line 292- delete (cross-sectional ref)

Line 294 intercounty

Lines 295-302:

This may be of use to see recommendations, to help guide practice and link to your findings

Duggan JD, Byrne PJ, Malone S, Cooper S-M, Moody J. High-Intensity Accelerations and Decelerations During Intercounty Camogie Match Play. Sports Health: A Multidisciplinary Approach. 2024;17(1):66-79. doi:10.1177/19417381241276016.

Line 303 – Reference (survey ref)

Line 303 educate the players?? How about the coaches?

Line 331-332 Game specific training?

Line 337-338 This seems just like a throw away statement, delete.

Lines 340-342 – This is concerning, did this not have a impact on data collection and analysis?

**Do you want your identity to be public for this peer review?** For information about this choice, including consent withdrawal, please see our Privacy Policy

Reviewer #1: **Yes:**  Dr John David Duggan

---

## [Decision Letter · Decision Letter 1]

18 Dec 2025

The Between Competition Sprint Profile of Intercounty Female Gaelic Football during Training and Match-Play: An Exploratory Study

PONE-D-25-51899R1

Dear Dr. McGuinness,

I am pleased to inform you that your manuscript has been judged scientifically suitable for publication and will be formally accepted for publication once it meets all outstanding technical requirements. I would like to thank you for the revisions and constructive corrections you have made to your manuscript to meet the requirements of the editor and reviewers. This can be an arduous process, so I thank the authorship team for their work here,

Have a good Christmas and all the best for the New Year 2026

Kind regards,

Shane Malone, Ph.D.

Academic Editor

PLOS One

Additional Editor Comments (optional):

Dear Authors,

Please feel open to choosing PLOSone again for your research should you feel that the paper meets the journal requirements.

Shane Malone, PhD

Reviewers' comments:

Reviewer's Responses to Questions

**Comments to the Author**

Reviewer #1: All comments have been addressed

2. Is the manuscript technically sound, and do the data support the conclusions?

Reviewer #1: Yes

3. Has the statistical analysis been performed appropriately and rigorously?

Reviewer #1: Yes

4. Have the authors made all data underlying the findings in their manuscript fully available?

Reviewer #1: Yes

5. Is the manuscript presented in an intelligible fashion and written in standard English?

Reviewer #1: Yes

Reviewer #1: Thank you for addressing the comments provided. They have enhanced the methodological and statistical rigor of the paper. They have also improved the practical applications for sports scientists and practitioners working in female Gaelic games.

**Do you want your identity to be public for this peer review?** For information about this choice, including consent withdrawal, please see our Privacy Policy

Reviewer #1: **Yes:**  Dr John David Duggan

---

## [Editor Report · Acceptance letter]

PONE-D-25-51899R1

PLOS One

Dear Dr. McGuinness,

I'm pleased to inform you that your manuscript has been deemed suitable for publication in PLOS One. Congratulations! Your manuscript is now being handed over to our production team.

Kind regards,

on behalf of

Dr. Shane Malone

Academic Editor

PLOS One